# Selective Voltammetric Detection of Ascorbic Acid from Rosa Canina on a Modified Graphene Oxide Paste Electrode by a Manganese(II) Complex

**DOI:** 10.3390/bios11090294

**Published:** 2021-08-25

**Authors:** Sophia Karastogianni, Dimitra Diamantidou, Stella Girousi

**Affiliations:** Analytical Chemistry Laboratory, Chemistry Department, Aristotle University of Thessaloniki, 54124 Thessaloniki, Greece; karastos@thea.auth.gr (S.K.); ddimitra93@gmail.com (D.D.)

**Keywords:** manganese(II) complex, modified graphene oxide paste electrode, ascorbic acid, square-wave voltammetry, rosa canina

## Abstract

Voltammetric techniques have been considered as an important analytical tool applied to the determination of trace concentrations of many biological molecules including ascorbic acid. In this paper, ascorbic acid was detected by square wave voltammetry, using graphene oxide paste as a working electrode, modified by a film of a manganese(II) complex compound. Various factors, such as the effect of pH, affecting the response characteristics of the modified electrode were investigated. The relationship between the peak height and ascorbic acid concentration within the modified working electrode was investigated, using the calibration graph. The equation of the calibration graph was found to be: I = 0.0550*γ*_ac_ + 0.155 with *R*^2^ = 0.9998, where I is the SWV current and γac is the mass concentration of ascorbic acid. The LOD and LOQ of the proposed method were determined to be 1.288 μg/L and 3.903 μg/L, respectively. Several compounds, such as riboflavin, biotin, and ions, such as Fe and Cu, were tested and it seemed that they did not interfere with the analytic signal. The proposed procedure was successfully applied in the determination of ascorbic acid in Rosa canina hips.

## 1. Introduction

Food is responsible for various diseases. Long ago these diseases were treated in empirical ways but from the beginning of the twentieth century onwards, their systematic study began. It turned out that these conditions were due to a lack of certain dietary factors called vitamins by Funk in 1912, thinking that they were all amines. However, after the isolation of the first vitamin in 1930, the above case turned out to be wrong but this name has prevailed to date [1]. Vitamins differ in their structure and function and are classified into two main categories, water-soluble and fat-soluble. Their most important role is to regulate the various reactions of metabolism. The lack of some vitamins breaks down special metabolic processes and can alter the metabolic balance of the body. Water-soluble vitamins are involved in the energy transfer and metabolism of protein, carbohydrates and fatty acids of cell membranes, while fat-soluble substances form an essential part of biological membranes and play an important role in maintaining their function and protecting them from oxidative damage or lipid peroxidation. Some vitamins act on the genetic material of the body and control the synthesis of certain enzymes [1].

L-Ascorbic acid (AA) also known as vitamin C is a water-soluble vitamin, mainly present in fruits and vegetables, animal feed, pharmaceutical formulations, multivitamin tablets and cosmetics [2,3]. Humans, along with other primates, guinea pigs, bats, birds, and other species, belong to the vertebrate class that cannot synthesize it on their own but it is a necessary nutrient for their diet [4]. The importance of AA in human health and wellbeing results from its role in several biological processes, such as hormone biosynthesis, amino acid metabolism, hydroxylation of collagen, free radical metabolism and immunity improvement. Thus, a lack of AA in the human body can cause various disorders, namely gastric irritations, anemia, poor wound healing and muscle degeneration [2,5].

High levels of ascorbic acid are associated with kidney problems and gastric irritation. Thus, it is often necessary to identify it in samples of food and beverages. Furthermore, AA can rapidly oxidize to dehydroascorbic acid (DHA) due to its low stability making the determination of this vitamin relatively difficult [6]. As a result, fast and sensitive analytical methods for the determination of AA are necessary for clinical, chemistry and diagnostics, and also for the food industry.

Recently, different analytical methodologies were reported, such as fluorescence [7], titration [8], high-performance liquid chromatography [9,10]. However, some of these techniques are time-consuming; some are poor in sensitivity and selectivity, while others are costly or need specially trained operators. On the other hand, electrochemical techniques offer the advantages of the low cost of buying and operating the instruments, the high sensitivity, and the simplicity of the method [11,12,13,14,15]. Furthermore, electroanalytical methods are reproducible, have relatively short analysis time and can be used in the direct analysis, without any extraction, clean-up or pre-concentration steps, and are easy to miniaturize and handle [11,12,13,14,15].

Stripping voltammetry is a useful tool that allows the electrochemical determination of many compounds including ascorbic acid. The working electrodes widely used in stripping voltammetry are solid carbon electrodes (CPE), vitreous carbon (GCE) and various metals (Pt, Au, Cu. Electrodes containing these mediators usually act as redox electrodes with respect to AA. In order to increase the sensitivity of the assumptions, the surface of the electrodes is formed with various materials and compounds, predominantly forming nanoscale electrodes. Characteristic examples are the formation of graphene oxide paste electrodes with Fe(III)-NClino nanomolar [16], [Cu(bp) (H_2_O)_2_]_n_ [17], and the modification of glassy carbon electrodes with gold nanoparticles [18]. Meanwhile, for the electrochemical determination of AA, electrodes with modified surfaces have been widely used, overlapping the fulling effect due to oxidation that may occur on the conventional electrode surfaces [6]. Carbon paste (CPE) [12,19] and glassy carbon (GCE) [20,21,22,23,24,25,26,27,28,29,30,31,32,33] modified electrodes are the most commonly used for AA analysis in the available literature. Moreover, Pt modified electrodes with 3,4-diaminobenzoic acid and neutral red [34] or screen-printed electrodes (SPE) [35,36,37]. Furthermore, boron-dropped electrodes were also used [38]. Meanwhile, miniaturized nanoporous gold (NPG) [39], platinum functionalized with commercial pre-polymers electrodes [40] and yttrium hexacyanoferrate microflowers on freestanding three-dimensional graphene substrates [41] were also utilized in the detection of AA.

Rosa canina is also called wild rose, roe, and canine. The canine works as a tonic in case of depletion and lack of vitamins, as well as a diuretic. Wildflower contains a large amount of vitamin C and vitamins A, B and K, gallic acid, antibiotics, tannin and flavonoid pigments (vitamin P), carotenoids (provitamin A), pectin, vegetable acids and essential oil. It is a very good herb for the skin and helps restore the disordered hormonal balance as well as in the irregular period. Its fruits are nutritious and act as mild laxatives, mild diuretics, and mildly astringent. It helps with infections and especially colds by improving the body’s defense. It helps in cases of constipation and mild gall bladder problems as well as in kidney and bladder diseases. The canines are prepared as a decoction that helps with colic problems, flatulence and as an antidiarrheal in dysentery. The fruit is noted for its high level of vitamin C and is used to make syrup, tea, and marmalade. It is grown or encouraged in the wild for the production of vitamin C from its fruit (often as rose-hip syrup), especially during conditions of scarcity or during wartime. Conclusively, these are some of the reasons it was chosen to be the real sample in the detection of ascorbic acid in our study.

For the determination of ascorbic acid in samples of wild rose fruit, the technique of high-pressure liquid chromatography [42], sometimes coupled to a mass spectrometer, is used [43,44]. However, the number of references that determine vitamin C in the particular substrate in the literature is limited [45,46] and was therefore chosen as a real sample in this work.

In this work, a modified GrOPE electrode was used to determine AA. The detection was based on immobilization of a mononuclear Mn (II) complex film on the electrode surface (Mn-GROPE), by applying differential pulse voltammetry (DPV). The proposed modified electrode was previously synthesized and characterized from our group, and was used in the determination of vitamin B_12_ with excellent analytical features [47,48]. This is the first time this electrode modifier was used in the detection of vitamin C, with also excellent analytical features, which are comparable and, in some cases, even better than those cited in the literature. The proposed sensor was found to have good selectivity against some interferents that are present in the Rosa canina sample. The reported method was then applied for the analysis of commercially available Rosa canina hips samples, using square wave voltammetry, which implies the applicability of the proposed sensor in real samples.

## 2. Experimental

### 2.1. Material and Methods

All reagents were of analytical grade unless stated otherwise and used as received. Dimethyl sulphoxide (DMSO), tetrahydrate manganese(II) chloride (MnCl_2_·4H_2_O) and ascorbic acid (vitamin C) were purchased from Merck (Darmstadt, Germany). Thiophene-2-carboxylic acid was purchased from Aldrich (Milwaukee, WI, USA). Triethanolamine and mineral oil were obtained from Sigma (Saint Louis, MO, USA). Graphene oxide was purchased from Sigma Aldrich (Saint Louis, MO, USA) (900704). (Mn(thiophen-2-carboxylic acid)2(triethanolamine)) [Mn(L)_2_(H_3_tea)] (**1**) was prepared as previously reported [49].

Stock solutions of 3 g L^–1^ of (**1**) were prepared after weighing a certain amount of the compound and diluting in dimethyl sulphoxide. All aqueous solutions were prepared with double-distilled water. Stock solutions of 3 g L^–1^ of ascorbic acid (vitamin C) were prepared after weighing a certain amount of the compound and dilution in double-distilled water and stored in the refrigerator for three days. Further diluted solutions of ascorbic acid (vitamin C) were made in double-distilled water before use. All the experiments were performed at ambient temperature in an electrochemical cell. The electrochemical cells were cleaned with diluted nitric acid and rinsed with double-distilled water. Ultrapure nitrogen was used to de-aerate the solutions by purging the dissolved oxygen for 15 min prior to each experiment.

Voltammetric experiments were carried out using a *μ*Autolab potentiostat/galvanostat and (Eco Chimie, Utrecht, The Netherlands) controlled by GPES 4.9.0005 Beta software. All electrochemical measurements were carried out at an ambient temperature, using a conventional three-electrode cell containing a platinum wire as a counter and Ag/AgCl/3 mol L^–1^ KCl electrode as reference electrodes, respectively. A graphene oxide paste electrode of 3 mm inner and 9 mm outer diameter of the PTFE sleeve was used as a working electrode. The pH of all solutions was measured using a Consort C830 pH meter (Consort bvba, Turnhout, Belgium).

### 2.2. Fabrication of the Electrochemical Sensor and Detection of Ascorbic Acid

The GrOPE was prepared by mixing by hand adequate amounts of GrOPE and paraffin oil of 75/25 mass ratio. A portion of the resulting mixture was packed into the bottom of the PTFE sleeve. The surface was polished to a smooth finish manually on a piece of weighing paper before use. Electrical contact was established via stainless steel screws. Subsequently, deposition was carried out. The determination of ascorbic acid (vitamin C) using standard solutions of a known concentration of ascorbic acid was carried out by the technique of adsorptive transfer stripping voltammetry. The proposed methodology followed the subsequent stages:

Fabrication of graphene oxide paste electrode (GrOPE).

Stirring for 300 s of the manganese complex dissolution solution (0.01 mol L^−1^ borate buffer pH 9.0 containing 0.02 mol L^−1^ KBr) by applying a potential of +0.0 V and using the differential pulse voltammetry (DPV), as well as scanning the electrode potential cathodically between +1.2 to 0.0 V to immobilize the complex on the surface of the GROPE (Mn-GrOPE).Then Mn-GrOPE was rinsed with double-distilled water and dried at room temperature.Stirring for 300 s in the dissolution solution of ascorbic acid (0.05 mol L^−1^ acetate buffer pH 6.8 containing 0.01 mol L^−1^ NaCl) without applying a potential for its adsorption to the manganese modified GrOPE (Vit C-Mn-GROPE).Afterward, the fabricated Vit C-Mn-GROPE was transferred in measurement solution (0.1 mol L^−1^ acetate buffer pH 5.4 containing 0.008 mol L^−1^ KBr), where the signal transduction took place, using square wave voltammetry and cathodically scanning the potential of the working electrode between +2.2 and +0.0 V, with a step potential equal to 0.0003 mV, a modulation amplitude equal to 0.08 mV and a frequency equal to 25 Hz. The resulting voltammogram in each measurement took into account the manganese reduction peak, which appeared at a potential of +0.7 V.

The raw data were treated using the Savitzky and Golay filter (level 2) of the GPES software, followed by the GPES software moving average baseline correction using a peak width of 0.03. Repeated measurements were carried out following the renewal of the GROPE surface by cutting and polishing the electrode.

### 2.3. Determination of Vitamin C in Real Samples

The Rosa canina sample was purchased from the market. Firstly, a small amount of rosehip was mashed in a porcelain capsule. After this portion being weighed, it was diluted 1:100 with double-distilled water and remained in the ultrasonic bath for about 10 min. Filtration with a 45 mm Millipore membrane and the second dilution of 1:25 then took place. For the determination of ascorbic acid, the standard addition method was followed. The solutions were prepared on the day of analysis to avoid the oxidation of ascorbic acid. The method followed for analyzing the real sample was that of adding known quantities of standard solution. In each of the five volumetric flasks containing the diluted amount of cyanide solution, a known volume of a standard ascorbic acid solution of 1 mg/L was added. No standard solution was added to the first (P_O_), 50 μL was added to the second (P_1_), in the third (P_2_) 100 μL, in the fourth (P_3_) 150 μL and finally in the fifth (P_4_) 200 μL. Then calibration curve was obtained from the measurements of the reduction signal of the standard solutions.

## 3. Results and Discussion

### 3.1. Electrochemical Behavior of the Modified Electrode Mn-GrOPE

The Mn complex with 2-thiophenecarboxylic acid and triethanolamine as ligands is first used to modify the electrode surfaces and for the first time, electrochemical film formation was studied on the surface of the graphene oxide paste electrode using a potentiostatic technique. The electrochemical behavior of the Mn-GrOPE was studied by cyclic voltammetry (CV) in 0.1 mol L^−1^ acetate buffer pH 4.6. The formed electrode was derived from 0.01 mol L^−1^ borate buffer containing 600 mg L^−1^ of Mn complex and 0.02 mol L^−1^ KBr after stirring for 300 s using the differential pulse voltammetry. Mn-GROPE was found to give two oxidation peaks at about +0.556 V and +0.854 V, respectively, and a reduction peak at about +0.355 V, which could be attributed to the oxidation and reduction of the complex’s central ion [50].

In addition, the effect of pH on the immobilized manganese complex film on the surface of GrOPE was studied. The formed film was transferred to buffers (4.0 ≤ pH ≤ 8.0) in the absence of the complex and the response of the reduction peak of the film was monitored. The reduction peak current response increases in the pH range from 4.0 to 4.6. Above the value of 4.6, the electrochemical response decreases by changing the pH to higher values. This signifies the possible involvement of hydroxyls in the reduction process. The maximum electrochemical response was observed when the pH equaled to 4.6. Interestingly, the change in the pH reduction potential of the complex film as a function of pH was found to decrease linearly by increasing the pH. It is reported that the slope of this diagram equals (ΔE/ΔpH) = −83.1 (±0.013) mV/pH. This value indicates the proton involvement in the film reduction reaction at GROPE and is close to the theoretical value of 88.8 mV/pH, predicted by the Nernst equation for an electrochemical action involving three protons and two electrons. Manganese complex(II) after its immobilization by applying +0.0 V for 300s and diffusion from bulk solution (pH = 9.0) to the GROPE/electrolyte interface was probably oxidized and adsorbed in the form of [Mn^4+^(L)_2_(tea)]^−^_ads_ onto electrode’s surface (Equations (1) and (2)) [49,50].
(1)$$[\mathrm{Mn}2^{+}(L{)}_{2}(H_{3}\mathrm{tea})]\;\to \;[\mathrm{Mn}2^{+}(L{)}_{2}(H_{3}\mathrm{tea}){]}_{\mathrm{surf}}$$
(2)$$[\mathrm{Mn}2^{+}(L{)}_{2}(H_{3}\mathrm{tea}){]}_{\mathrm{surf}}\;\to \;[\mathrm{Mn}4^{+}(L{)}_{2}(\mathrm{tea}){{]}^{-}}_{\mathrm{ads}}+3H^{+}+2e^{-}$$

Thus, and based on our previous studies [49,50], by differentially pulse scanning the potential cathodically between +1.2 to +0.0 V the reduction reaction probably follows the subsequent total Equation (3). It must be stressed, though, that more experiments should be performed to confirm this conclusion.
[Mn^4+^(L)_2_(tea)]^−^_ads_ + 3H^+^ +2e^−^ → [Mn^2+^(L)_2_(H_3_tea)]_ads_(3)

In order to confirm whether the film is in a polymer or monomeric form, its electrochemical response was studied using cyclic voltammetry, Figure 1. Figure 1 shows the cyclic voltammograms of the film formed on GrOPE, when 0.1 mol L^−1^ a buffer solution with pH 9.0 was used and by scanning the potential between +0.0 to 1.2 V for 15 scans. The results show that at pH 9.0 no oxidation and reduction peaks appear. Furthermore, as the number of scan cycles increases the current response decreases progressively by increasing the number of scan cycles [49,50]. This means that it is most unlikely that the polymerization of (1) occurred under the studied conditions. Meanwhile, since there is a large amount of [Mn^2+^(L)_2_(H_3_tea)]_ads_ species, the [Mn^2+^(L)_2_(H_3_tea)]_ads_ diffusion rate is larger than the electron transfer rate from the [Mn^2+^(L)_2_(H_3_tea)]_ads_ to the electrode, so that the oxidation current does not saturate in the present potential range [51,52]. Moreover, it is likely that some product was formed on the electrode surface during the scanning. This product, being very thin and almost transparent, prevents the monomer from polymerizing [51,52].

The electrochemical behavior of GrOPE and Mn-GrOPE in a solution of 1 × 10^−3^ mol L^−1^ K_3_[Fe(CN)_6_] was also investigated by means of CV, Figure 2. As a carrier electrolyte, a solution of 0.2 mol L^−1^ KCl was used. The potential was scanned in the range from −0.6 V to + 0.6V with a scan rate of 0.005 Vs^−1^. As can be seen, the GrOPE electrode (Curve 1) shows a pair of defined peaks with an upward potential of +0.132 V and the down-peak potential at −0.007 V. Using the Mn-GrOPE electrode (Curve 2, Figure 2) the current intensities of the respective peaks increased, while the value of the potential difference between the anodic and cathodic peak was reduced from 125 mV to 119 mV. This may be due to the greater electrochemical activity of Mn-GrOPE. The increase in the potential difference between the anodic and the cathodic peak suggests the existence of a negatively charged surface layer of the electrode surface, which can attract positively charged species. An increase in peak current intensity indicates the largest electrochemical activity of the formed electrode resulting from its largest electroactive surface.
*Ip* = 2.60 × 10^5^ *C* A *D*^1/2^ *n*^2/3^ *u*^1/2^(4)

With the help of the Randles–Sevcik equation (Equation (4)), the electroactive surface area (*A*) of the GrOPE electrode was determined by replacing the values of *D*, *n*, *u* and *C*. For the redox system studied (K_3_[Fe(CN)_6_]), *n* = 1, *D* = 7.6 × 10^−6^ cm^2^ s^−1^. Based on the above, the electroactive surface area of the GrOPE was calculated to be 0.022 cm^2^. In addition, the dependence of the current intensity on the square root of the scan rate is linear for the anodic and cathodic peaks in the case of Mn-GrOPE. The slope of this diagram equals Δ*Ι*/Δ*u*^1/2^ = 5 × 10^−5^ A s V^−1^ for both peaks. Therefore, this value will be used to calculate the electroactive surface area (*A*) of the Mn-GrOPE electrode. Thus, with the help of the Randles–Sevcik equation (Equation (4)), the electroactive surface area (*A*) of the Mn-GrOPE electrode was determined. The ratio of the peak current to the square root of the scan rate of the potential (*I*/*u*^1/2^) was calculated from the slope of the oxidation curve. Based on the above, the electroactive surface area of the Mn-GrOPE was calculated to be 0.070 cm^2^. By comparing the results for the two electrodes, the use of the manganese complex as a surface modifier increases the electroactive surface area and enhances the response of the K_3_[Fe(CN)_6_] system.

### 3.2. Morphology of the Modified Mn-GrOPE

The morphological characteristics of GrOPE and Mn-GrOPE electrodes were studied by scanning electron microscopy (SEM). From Figure 3 it is clear that there are morphological differences between the GrOPE electrode and Mn-GrOPE, which confirm the deposition of the manganese complex on the surface of the GrOPE. The GrOPE electrode (Figure 3a) is characterized by irregularly shaped graphite beads with considerable consistency and small pore size. In the Mn-GrOPE electrode, the complex film is homogeneously distributed in the mineral oil and additionally has a sufficiently compact structure, as is in GrOPE (Figure 3b–d), but has larger pores (Figure 3b,d). Furthermore, as shown in Figure 3b,c, the film particles are irregularly sized and different aggregates on the GrOPE, which are arranged in tamped tiles on the electrode surface. Figure 3b–d show that these agglomerates have a fairly high roughness. The white beads in the miniaturization (Figure 3b,d) can be attributed to the growth of the particles of the manganese complex film in the GrOPE. However, a closer look at the miniature of Figure 3 reveals that small structures of spherical shape also exist.

### 3.3. Electrochemical Behavior of Vitamin C on the Modified Mn-GrOPE

Meanwhile, the electrochemical behavior of ascorbic acid was studied in GrOPE and Mn-GrOPE in 0.1 mol L^−1^ acetate buffer pH 5.4 with CV, Figure 4A. The results showed that two oxidation peaks appeared in the case of Mn-GrOPE at about +0.744 and +0.892 V vs. Ag/AgCl and a reduction peak at +0.494 V vs. Ag/AgCl, due to the oxidation of the Mn^2+^ central metal ion (Figure 3A, curve 2). On the other hand, only a peak occurred in the case of the oxidation of ascorbic acid on GrOPE at about +0.363 V, while the reduction peaks were absent from the voltammograms (Figure 4A curve 3). Finally, two oxidation peaks occurred in the case of oxidation of ascorbic acid in Mn-GROPE at approximately +0.309 and +0.809 mV vs. Ag/AgCl (Figure 4A curve 4), due to the oxidation of ascorbic acid and the central metal ion Mn^2+^, respectively. In the reverse direction, a peak of about +0.518 mV (Figure 4A curve 4) appeared, due to the reduction of manganese(II) complex [50]. It is worth noting that no reduction peak corresponding to ascorbic acid appeared. Using the Mn-GrOPE electrode the intensity of the ascorbic acid oxidation peak current increased (comparison of curves 3 and 4 of Figure 4A), while the current intensity Mn-GrOPE reduction peak remained almost constant in the presence of ascorbic acid. The potential of the oxidation peak of ascorbic acid and Mn-GrOPE was shifted to negative values, whereas the peak potential of Mn-GrOPE remained virtually unchanged in the presence of ascorbic acid. This may be due to the greater electrochemical activity of Mn-GrOPE compared to that of GrOPE and that the electron transfer of AA was irreversible in both cases. This conclusion is confirmed by the larger electroactive surface area and Mn-GrOPE, as calculated in the previous paragraph.

In addition, it was found that the oxidation peak current of ascorbic acid in Mn-GrOPE increased linearly by increasing the scan rate of the potential, indicating its adsorption to the surface of Mn-GrOPE. On the other hand, the ascorbic acid oxidation peak current in Mn-GrOPE was found to be linearly dependent on the square root of the potential scan rate, suggesting that diffusion was also involved in the electrochemical process. It should be noted that no reduction peaks were present in CVs, which confirms the irreversibility of the electrochemical process.

Meanwhile, Figure 4C gives the voltammograms of the reduction of GrOPE (curve a), ascorbic acid on GrOPE (curve b), Mn-GrOPE (curve c) and ascorbic acid on Mn-GrOPE. The results showed that ascorbic acid is not reduced in both GrOPE and Mn-GROPE (curves b and d of the Figure 3C, respectively) in the range of +1.4 V to −0.6 V. Followingly, from Figure 4C it is derived that the presence of ascorbic acid resulted in a rapid increase in the peak intensity of the Mn-GrOPE reduction peak at +0.548 V. Furthermore, the shifting of the potential to more negative values relative to that of Mn-GrOPE in the absence of ascorbic acid (about 73 mV) (comparison of curves c and d) indicates the potential interaction of ascorbic acid with the proposed electrode.

Interestingly, the change in the pH reduction potential of the complex film after its interaction with vitamin C as a function of pH was found to decrease linearly by increasing the pH. It is reported that the slope of this diagram equals (ΔE/ΔpH) = −60.5 (±0.018) mV/pH. This value indicates the proton involvement in the film reduction reaction at GROPE and is close to the theoretical value of 59.2 mV/pH, predicted by the Nernst equation for an electrochemical action involving two protons and two electrons.

The electrochemical behavior of ascorbic acid was studied in GrOPE and Mn-GrOPE in 0.1 mol L^−1^ acetate buffer pH 5.4 with SWV. Figure 4B gives the voltammograms of the oxidation of GrOPE (curve a), Mn-GrOPE (curve b), ascorbic acid on GrOPE (curve c) and ascorbic acid on Mn-GrOPE (curve d). An oxidation peak at about +0.787 V appeared, due to the oxidation of the central metal ion Mn^2+^ to Mn^4+^ [50]. From the curve c of Figure 4B it appears that when GrOPE is used, the presence of ascorbic acid causes a peak at +0.216 V. In the presence of ascorbic acid, a peak appears in the voltammograms when the electrode used is Mn-GrOPE at about +0.489 V (curve d in Figure 4B). This peak is probably due either to the oxidation of the central metal ion Mn^2+^ or to the oxidation of ascorbic acid. In addition, the peak current intensity is reduced (comparison of curves b, c and d). Thus, the use of Mn-GrOPE probably worsens the oxidation signal of ascorbic acid.

Based on these findings and the literature [51,52,53,54,55,56], the electrochemical reaction mechanism of AA on Mn-GrOPE could be seen in Figure 5. Therefore, the AA molecules in the solution were adsorbed onto the surface of Mn-GrOPE. Then the AA molecules hydrolyzed and were oxidized to dehydroascorbic acid. The oxidation process of AA.

In conclusion, the presence of ascorbic acid improves the response of the Mn-GrOPE reduction signal. By comparing the figures, we observe that the presence of Mn-GrOPE aggravates the oxidation signal of ascorbic acid during oxidation, whereas the presence of ascorbate significantly improves the response of the Mn-GrOPE reduction signal. Thus, it was chosen to detect ascorbic acid through the Mn-GrOPE reduction peak.

### 3.4. Optimization of Experimental Conditions

The interaction potential of ascorbic acid with Mn-GrOPE was studied in order for the proposed vitamin detection assay on Mn-GrOPE to have the best response after the interaction with the vitamin. The experiments were carried out to optimize the ascorbic acid preconcentration potential on the Mn-GrOPE. The results showed that this parameter had no effect on the signal. Therefore, it was decided not to apply potential at the stage of vitamin preconcentration.

Then, the interaction time of ascorbic acid with Mn-GrOPE, Figure 6, was studied. Interaction time is one of the most important factors influencing the sensitivity of the method. It determines the degree of signal increase while at the same time affects the frequency of analysis, considering that the preconcentration stage is the most time-consuming stage of the analysis. The study of the effect of the interaction time of ascorbic acid with the modified Mn (II) graphene oxide paste electrode was in a range of 30 s to 600 s, Figure 5. From Figure 6, it is observed that by increasing the concentration-time, the current increases to a point, stabilizes for a short time and then decreases. Consequently, it is concluded that the method is best applied to times between 300 and 400 s, where the signal is higher, and therefore, as an interaction time, the value of 300 s was chosen, bearing in mind the time of the analysis.

## 4. Analytical Performance of the Proposed Assay

SWVs after the interaction of different mass concentrations of ascorbic acid on Mn-GrOPE are shown in Figure 7. Standard solutions were prepared by appropriately diluting a standard solution of 0.05 mg L^−1^ of ascorbic acid and analyzed by the proposed method. The calibration curve is shown in Figure 8. For each standard solution, five measurements were taken and the average was calculated.

The calibration graph obtained (Figure 8) showed a good linear relationship between the SWV peak current and the AA concentration from 3.903 (corresponding to 2.22 × 10^−8^ mol L^−1^) to 158 (corresponding to 8.97 × 10^−10^ mol L^−1^) μg L^−1^ with a correlation coefficient of 0.9998. The detection limit was calculated using 3 sb/slope, where sb is the standard deviation of the blank measurements and the slope of the calibration curve. A detection limit of 1.228 μg L^−1^ (corresponding to 6.97 × 10^−9^ mol L^−1^) was obtained.

Then the regression equation and the linear correlation coefficient *R* were calculated (Equation (5)), for a preconcentration time of 300 s.
*A* (μA) = 0.0550 (±0.0010) *γ*_AA_ (μg L^−1^) + 0.155 (±0.022), *R* = 0.9998(5)
where *A*: the measured value of the Mn-GROPE reduction peak current and *γ*_AA_: the mass concentration of ascorbic acid.

The relative standard deviation was measured at two levels of concentration mass of AA, i.e., 45 μg L^−1^ and 158 μg L ^−1^ and calculated to be 4.2 % and 3.8%, respectively, values indicating the good reproducibility of the proposed assay. The linear detection region of AA in the present study is in several cases better than that determined by the corresponding electrochemical techniques in the literature [11,12,13,14,15,16,17,18,19,20,21,22,23,24,25,26,27,28,29,30,31,32,33,34,35,36,37,38,39,40,41,45] On the other hand, the detection limit of the current analytical AA detection methodology is better compared to the majority of those determined with the other corresponding electrochemical techniques available in the literature, with the exception that it is in the same magnitude of Zhao et 2019 [20], Kaçar and Erden 2020 [2], Broncová et al. 2021 [34], Pekin et al. [45], as it can be seen from Table 1.

## 5. Selectivity of the Proposed Assay

The selectivity of the sensor was investigated (Table 2) by using some possible interfering substances. Tolerance of analysis to species that potentially interfere with a specific study is tested by analyzing mixtures of these species with the analyte at a mass concentration ratio of 100:1. Recovery rates of 100 μg L^−1^ vitamin C in the 95–105% range were set as the acceptance criterion of the method.

Indicatively, two series of measurements were performed for vitamin C using the proposed methodology, one referring to the electrochemical response of the inoculated sample to the inhibitor and the other to measure the electrochemical response of vitamin C alone. The results showed that the substances studied did not interfere in the determination of vitamin C. Thus, the use of the Mn-GrOPE electrode significantly improves the selectivity and consequently the applicability of the proposed method of determining vitamin C. Other possible inhibitors of retinol palmate (vitamin A) and tocopherol (vitamin E) could not prevent identification due to their insolubility in aqueous solutions.

## 6. Determination of Vitamin C in Real Samples

The proposed methodology was applied to the determination of vitamin C in an extract from a commercially available Rosa canina sample by applying the method of adding known quantities of standard solutions. Figures a and b give the curves of the addition of known quantities of standard solutions. The regression equation for the was found to be equal to *I*(A) = (5.83 × 10^−5^) × (mL) + 1.51 × 10^−6^ with a linear correlation coefficient *R* = 0.9998.

According to the results, the amount of vitamin C was determined to be equal to (4.84 × 10^−6^ ± 0.003) g L^−1^. The analytical results are in good agreement with the bibliographic values determined by other methods [18]. For example, Vlasta Cunja et al. [39] determined the concentration of ascorbic acid in Rosa canina to be equal to 716.8 mg/100 g. It should be noted that the results were obtained using five different electrodes in each sample and each measurement was repeated five times.

## 7. Conclusions

In this paper, a modified graphene oxide paste electrode with [Mn (L)_2_ (H_3_tea)] was made and characterized and the results showed that the presence of the complex film significantly improved the characteristics of the electrode. The modified electrode produced square wave signals depending on the vitamin C mass concentration in the range 3.903–158 μg L^−1^. The methodology applied proved to be simple, fast and economical with a fairly low detection limit. It exhibits very good selectivity, as the compounds with which Vitamin C is likely to coexist do not interfere with its determination. Finally, the method can be applied to real samples successfully.

## Figures and Tables

**Figure 1 biosensors-11-00294-f001:**
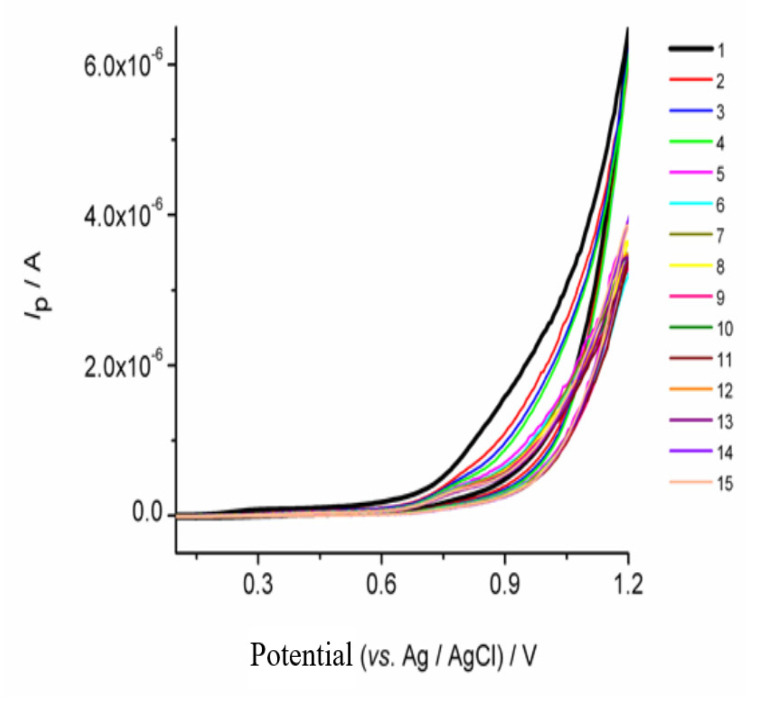
Successive cyclic voltammograms of 600 mg L^−1^ of (1) in 0.1 mol L^−1^ phosphate buffer pH = 9.0: (1) number of scan cycle = 1, (2) number of scan cycle = 2, (3) number of scan cycle = 3, (4) number scan cycle = 4, (5) number of scan cycle = 5, (6) number of scan cycle = 6, (7) number of scan cycle = 7, (8) number of scan cycle = 8, (9) number of scan cycle = 9, (10) number of scan cycle = 10, (11) number of scan cycle = 11, (12) number scan cycle = 12, (13) number of scan cycle = 13, (14) number of scan cycle = 14 and (15) number of scan cycle = 15 (scan rate potential = 0.01 V s^−1^ and step potential = 0.006 V).

**Figure 2 biosensors-11-00294-f002:**
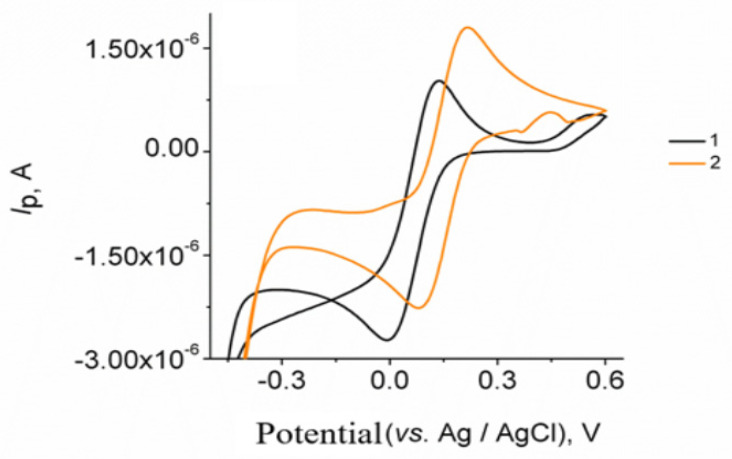
Cyclic voltammograms of 1 × 10^−3^ mol L^−1^ K_3_[Fe(CN)_6_] (1) GrOPE electrode, (2) Mn-GrOPE electrode, ((600 mg L^−1^ of [Mn(L)_2_(H_3_tea)], other experimental conditions are given in the experimental section).

**Figure 3 biosensors-11-00294-f003:**
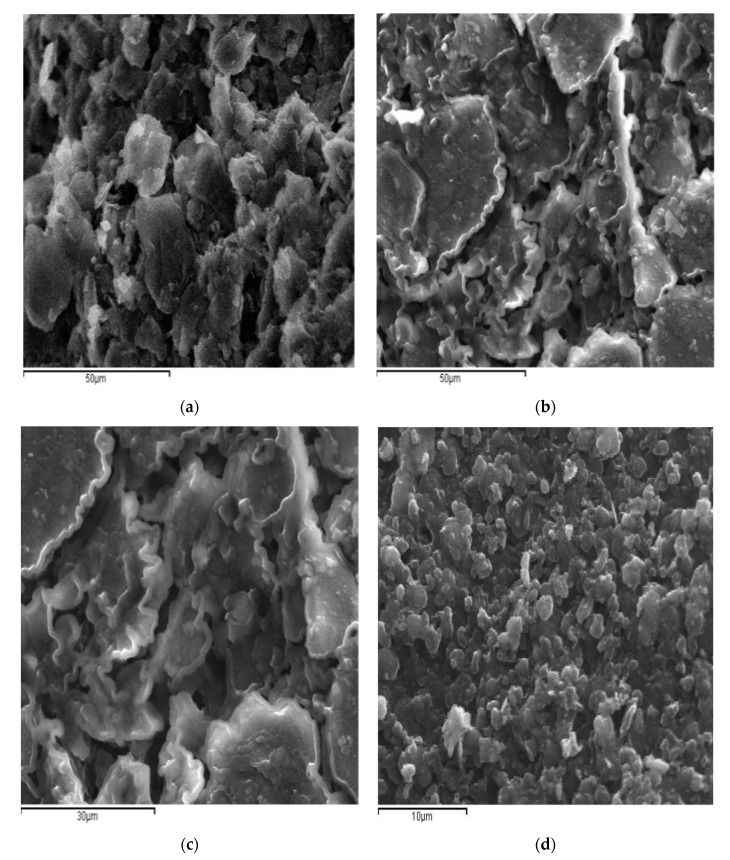
SEM images for electrodes (**a**) GrOPE, (**b**–**d**) Mn-GrOPE at different magnifications.

**Figure 4 biosensors-11-00294-f004:**
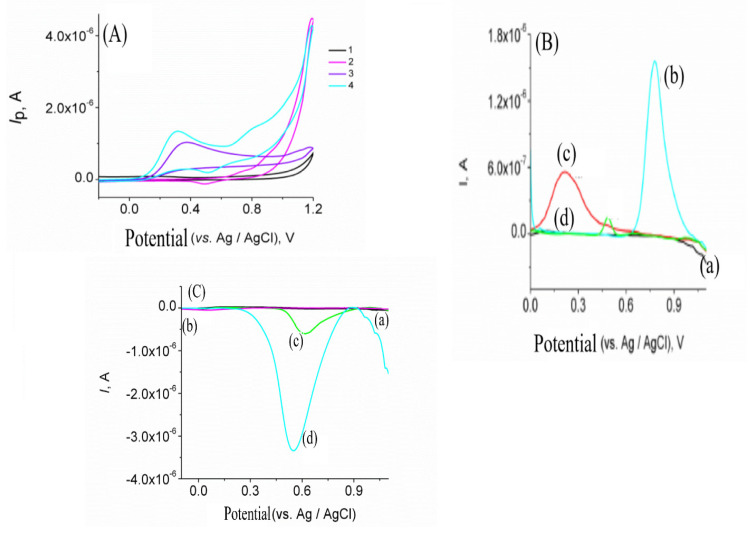
(**A**) Cyclic voltammograms of GrOPE (curve 1), Mn-GrOPE (curve 2), ascorbic acid on GrOPE (curve 3) and ascorbic acid on Mn-GrOPE (curve 4) (conditions: start potential = first vertex potential = +0.6 V, second vertex potential = +1.2 V, l step potential = 0.006, scan rate = 10 mV s^−1^ and number of scans = 1). (**B**) Square wave oxidation voltammograms of GrOPE (curve a), Mn-GrOPE (curve b), ascorbic acid on GrOPE (curve c) and ascorbic acid on Mn-GrOPE (curve d), (**C**) Square wave reduction voltammograms of GrOPE (curve a), ascorbic acid on GrOPE (curve b), Mn-GrOPE (curve c) and ascorbic acid on Mn-GrOPE (curve d) (other experimental conditions as mentioned in the experimental section).

**Figure 5 biosensors-11-00294-f005:**
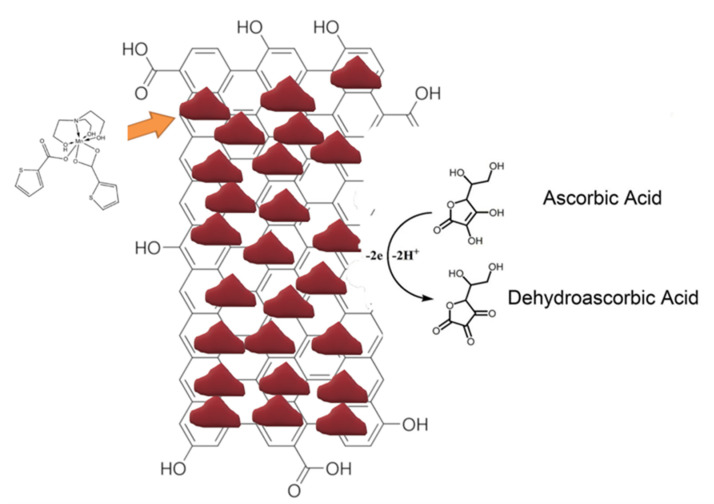
Schematic representation of possible AA electrochemical sensing mechanism by using Mn-GrOPE electrode.

**Figure 6 biosensors-11-00294-f006:**
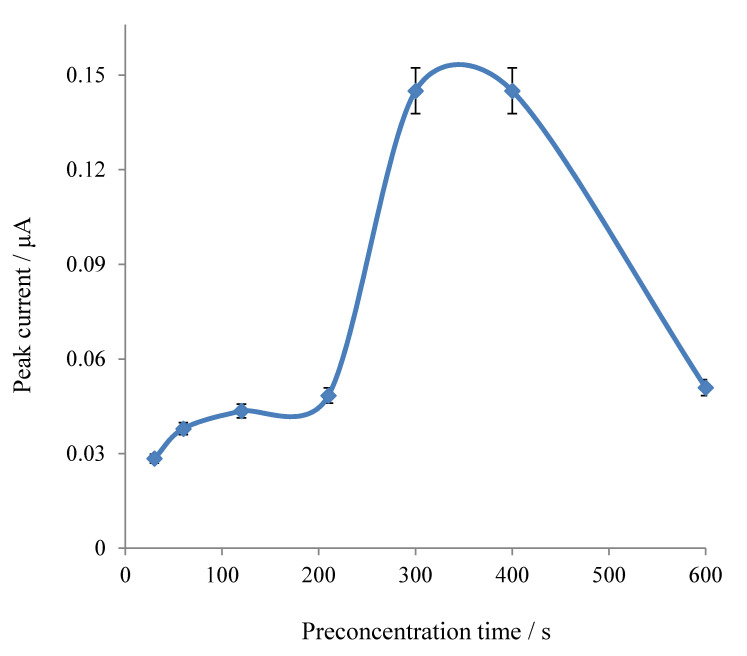
Effect of ascorbic acid preconcentration time on Mn-GrOPE (reduction peak current is given in absolute values).

**Figure 7 biosensors-11-00294-f007:**
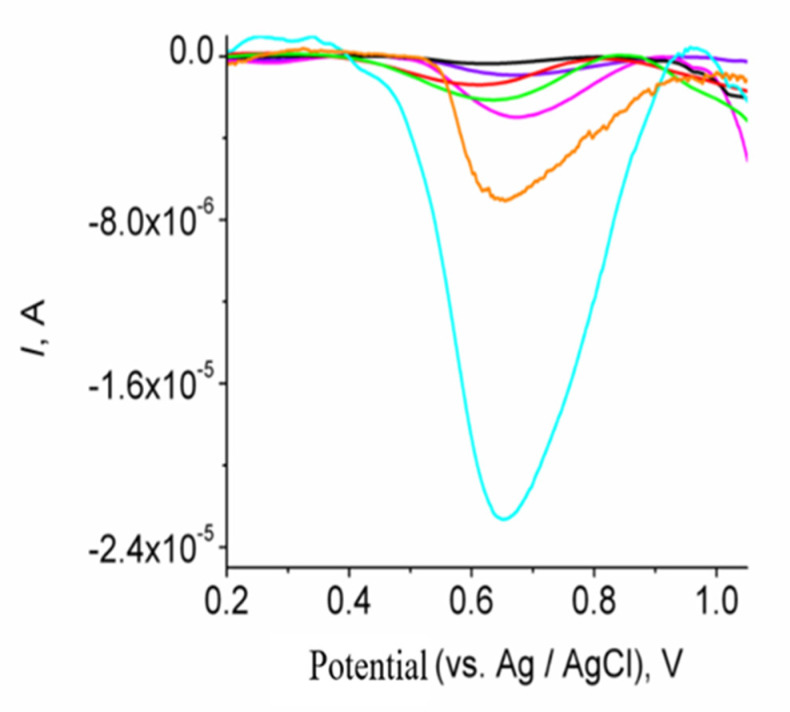
Reduction SWVs of Mn-GrOPE after its interaction with ascorbic acid of different mass concentrations in the selected conditions (range of ascorbic acid mass concentrations: 1.288–3.903 μg L^−1^ and other experimental as mentioned in the experimental section).

**Figure 8 biosensors-11-00294-f008:**
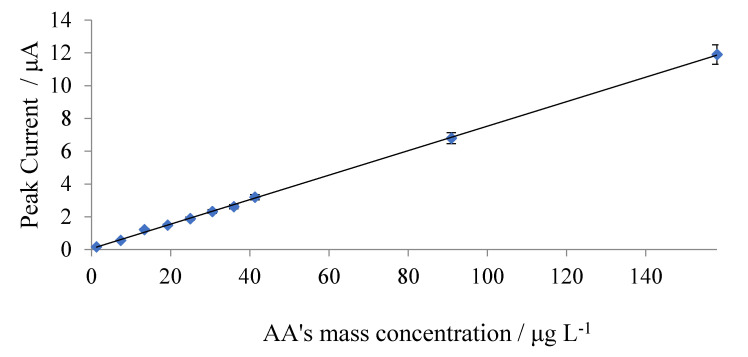
Calibration curve for the determination of ascorbic.

**Table 1 biosensors-11-00294-t001:** Analytical features of selected electrochemical sensors for determination AA.

Analytes	Electrode	Modifier	Detection Limit	Linear Range	Sample	Analytical Technique	Reference
Ascorbic acid	GCE	CNFs, NDs, and P(L-Asp)	0.1 μM	0.2–1800 μM	Vitamin C and effervescent tablet and pharmaceutical powder	CV	[2]
Ascorbic acid	Au-gr/CVE	Carbon veil (CV) and phytosynthesized gold nanoparticles (Au-gr)	0.05 μM	1 M–5.75 mM	Fruit juice	Cyclic and linear sweep voltammetry, chronoamperometry	[11]
Ascorbic acid	CPE	MIL−101	6 mM	0.01–10 mM	Pharmaceuticals	CV, EIS, and SWV	[12]
Ascorbic acid	GCE	Graphene oxide, multi-walled carbon nanotubes and gold nanorods	0.85 nM	1 nM–0.5 μM and 1 μM–0.8 mM	Serum	Cyclic voltammetry (CV), Differential Pulse Voltammetry (DPV)	[20]
Ascorbic acid, dopamine and uric acid	GCE	Porous g-C_3_N_4_ assembled with graphene oxide (GO)	3.7–39 μM	5–1300 μM	Serum	CV, DPV	[21]
Ascorbic acid, dopamine and uric acid	GCE	poly(3,4-ethylenedioxythiophene) (PEDOT) and polyaniline (PANI)	24.2 μM	100 to 10,000 μM	Serum	CV	[22]
Ascorbic acid	GCE	Silver nanoparticles (AgNPs): Polyvinylpyrrolidone (PVP)	0.047 μM	0.2–1200 μM	Fruits	DPV	[23]
Epinephrine, uric acid and ascorbic acid	GCE	-	0.5 mg/L	4.98–578.95 mg/L	Dietary supplement	SWV	[24]
Ascorbic acid, dopamine and uric acid	GCE	Polyvinylpyrrolidone (PVP)—GR	0.8 μM	4.0 μM–1.0 mM	Urine	CV	[25]
Ascorbic acid	GCE	Black phosphorus nanosheets (BPNS)	0.3 nM	1–35 nM	-	DPV	[28]
Ascorbic acid	Pt	4-amino−2,1,3-benzothiadiazole, 3,4-diaminobenzoic acid, and neutral red	-	-	Tablets	CV, columetric titration	[34]
Ascorbic and oxalic acids	SPE, GCE	Au and Pd particles		1.0 × 10^−8^ to 5.0 × 10^–3^ M	juices and fruits	CV	[36]
Acetaminophen, Ascorbic acid and Uric acid	multi screen-printed electrode	-	-	-	-	CV	[37]
Ascorbic acid	Boron doped diamond electrode (BDD)	-	1.87 μM	-	Tablets	CV, Square Wave Voltammetry (SWV)	[38]
Ascorbic acid	CPE	Sepiolite clay (SC) nanoparticles	4.2 × 10^−9^ M	1.4 × 10^−8^–9.0 × 10^−7^ M	Pharmaceutical formulations but also natural products such as vitamin C-rich fruit Rosa canina and mineral waters	-	[45]

^a^ GCE: glassy carbon electrode, ^b^ CNFs: carbon nanofibers, ^c^ NDs: nanodiamonds, ^d^ P(L-Asp): poly(L-aspartic acid), ^e^ CV: cyclic voltammetry, ^f^ Au-gr/CVE: phytosynthesized gold nanoparticles (Au-gr) modified carbon veil electrode, ^g^ CPE: carbon paste electrode, ^h^ MIL−101-(Cr): Metal-Organic Framework MIL-101-(Cr), ^i^ EIS: electrochemical impedance spectroscopy, kSWV: square wave voltammetry.

**Table 2 biosensors-11-00294-t002:** Interference study of the proposed vitamin C detection assay.

Interferents	Recovery/%
Riboflavin	102.0
Biotin	103.59
Pyridoxal	100.9
Niacin	104.4
Pantothenic acid	102.3
Thiamin	100.7
Folic acid	99.8
Caffeic acid	102.7
Gallic acid	103.5
Glucose	98.4
Fructose	98.7
Fe	102.0
Cu	101.0
Al	99.8
Zn	103.7
Mg	98.0
Ni	99.3

## Data Availability

The data presented in this study are available on request from the corresponding author.

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
