# Peer review of "Selective Voltammetric Detection of Ascorbic Acid from Rosa Canina on a Modified Graphene Oxide Paste Electrode by a Manganese(II) Complex"

_biosensors, 2021, doi:10.3390/bios11090294_

Round 1
Reviewer 1 Report
the main points that should be corrected were listed in the sent review in detail.
The main repairs should be carried out in
*** "Introduction": to focus on the ascorbic acid determination and the innovation that their procedure offers
*** "Experimental part": to specify whether it is a monomeric or polymeric form of the receptor.
*** "Discussion and results":
i) in the case of a polymeric form, it is necessary to characterize the receptor deposited on the electrode surface by suitable spectroscopic techniques (Infrared, raman spectroscopy)
ii) to explain the mechanism of the response to ascorbic acid formation and the role of the receptor in its detection.
Author Response
*** "Introduction": to focus on the ascorbic acid determination and the innovation that their procedure offers.
It is focused, see lines 67-94, lines 110-114 and lines 120-126.
*** "Experimental part": to specify whether it is a monomeric or polymeric form of the receptor.
The term. It was corrected, see page 4 lines 116-117.
*** "Discussion and results":
- in the case of a polymeric form, it is necessary to characterize the receptor deposited on the electrode surface by suitable spectroscopic techniques (Infrared, raman spectroscopy)
We don’t have these means to identify it. We assume that it is probably, see page 8-9 lines 225-257.
- to explain the mechanism of the response to ascorbic acid formation and the role of the receptor in its detection.
see lines 314-393, 3.3 Electrochemical behavior of vitamin C on the modified Mn-GrOPE.
Reviewer 2 Report
Dear All
The present manuscript reported that Selective voltammetric detection of ascorbic acid (vitamin C) from Rasa canina on a modified graphene oxide paste electrode by a manganese(II) complex. It needs to improve, please find my comments, please answer all comments carefully.
- Title //Please use “ascorbic acid or Vitamin C” in the title.
- in the text: Rosa and in title: Rasa !!!
- Why did you choose “ Rosa canina hips “ ? please explain?
- Introduction, please modify and improve it, page 2. Line 45,
“In pure form is a crystalline white solid, while when contaminated or impurities it appears slightly yellow. It is soluble in water and its solutions are slightly acidic. The name "ascorbic acid" derives from the prefix a- and scurvy, a hemorrhagic dis-ease known since antiquity, due to the lack of vitamin C. Vitamin C is a glucose derivative and many animal organisms can produce it on their own”
This information is really elementary, ascorbic acid is crystalline white solid and etc. please remove them. In the whole introduction part.
- page 2, line 63, the weak part of your paper
“Recently, different analytical methodologies have been reported, such as fluorescence [7], titration [8], high performance liquid chromatography [9, 10].”
Please add more information, electroanalysis is one of the main choices for the analysis of ascorbic acid.
Please add suitable references:
---Freeman, Christopher J., Borkat Ullah, Md Islam, and Maryanne M. Collinson. "Potentiometric Biosensing of Ascorbic Acid, Uric Acid, and Cysteine in Microliter Volumes Using Miniaturized Nanoporous Gold Electrodes." Biosensors 11, no. 1 (2021): 10.
---Broncová, G., Prokopec, V., & Shishkanova, T. V. (2021). Potentiometric Electronic Tongue for Pharmaceutical Analytics: Determination of Ascorbic Acid Based on Electropolymerized Films. Chemosensors, 9(5), 110.
----Monti, P., Migheli, Q., Bartiromo, A. R., Pauciulo, A., Gliubizzi, R., Marceddu, S., ... & Delogu, G. (2019). A Storage-Dependent Platinum Functionalization with a Commercial Pre-Polymer Useful for Hydrogen Peroxide and Ascorbic Acid Detection. Sensors, 19(11), 2435.
--- Hatamie, A., Rahmati, R., Rezvani, E., Angizi, S., & Simchi, A. (2019). Yttrium hexacyanoferrate microflowers on freestanding three-dimensional graphene substrates for ascorbic acid detection. ACS Applied Nano Materials, 2(4), 2212-2221.
- please add more information: “electropolymerization of a mononuclear Mn (II) complex” about the mechanism? reactions?
- page 10, line 370, “ Recovery rates of 100 ng L−1 vitamin C in the 95-105 % range” It is the very low concentration that has been used for recovery studies, please check the concentration units, linear range? it is not smaller than the linear range. Please see, Page 11, line 387, regression equation, and page 11, line 400.
- Figure 4 and Figure 5, the current intensity is negative and then is positive ?!
- please add more information about electrocatalytic ascorbic acid detection at the modified electrodes. and interaction with ascorbic acid of electrode modifier. Figure 5,
- page 5, line 219: “geometric area”
I do not agree with you about the calculation of geometric area of the electrode, in fact, in the purposed electrochemical method you can measure electroactive surface area, please check the "Bard books" and related references.
the geometric area is a physical area; you can see it. but the electrochemical reaction does not occur on all physical surfaces. for this reason, the estimated area is lower than the physical area.
- page 4, line 154, “ (0.1 mol L−1 acetate buffer pH 5.4 containing 0.008 mol L−1 KBr),” why you added KBr to buffer ???!
Author Response
- Title //Please use “ascorbic acid or Vitamin C” in the title.
It was corrected, see page 1 line 1.
- in the text: Rosa and in title: Rasa !!!
It was corrected see page 1 line 2.
- Why did you choose “ Rosa canina hips “ ? please explain?
see page 4 lines 95-109 and 110-114.
- Introduction, please modify and improve it, page 2. Line 45,
It was removed, see page 2.
“In pure form is a crystalline white solid, while when contaminated or impurities it appears slightly yellow. It is soluble in water and its solutions are slightly acidic. The name "ascorbic acid" derives from the prefix a- and scurvy, a hemorrhagic dis-ease known since antiquity, due to the lack of vitamin C. Vitamin C is a glucose derivative and many animal organisms can produce it on their own”
This information is really elementary, ascorbic acid is crystalline white solid and etc. please remove them. In the whole introduction part.
It was removed, see page 2.
- page 2, line 63, the weak part of your paper
It was corrected, see page 3 lines 63-64.
“Recently, different analytical methodologies have been reported, such as fluorescence [7], titration [8], high performance liquid chromatography [9, 10].”
Please add more information, electroanalysis is one of the main choices for the analysis of ascorbic acid.
It was added, see page 3 lines 72-75.
Please add suitable references:
---Freeman, Christopher J., Borkat Ullah, Md Islam, and Maryanne M. Collinson. "Potentiometric Biosensing of Ascorbic Acid, Uric Acid, and Cysteine in Microliter Volumes Using Miniaturized Nanoporous Gold Electrodes." Biosensors 11, no. 1 (2021): 10.
---Broncová, G., Prokopec, V., & Shishkanova, T. V. (2021). Potentiometric Electronic Tongue for Pharmaceutical Analytics: Determination of Ascorbic Acid Based on Electropolymerized Films. Chemosensors, 9(5), 110.
----Monti, P., Migheli, Q., Bartiromo, A. R., Pauciulo, A., Gliubizzi, R., Marceddu, S., ... & Delogu, G. (2019). A Storage-Dependent Platinum Functionalization with a Commercial Pre-Polymer Useful for Hydrogen Peroxide and Ascorbic Acid Detection. Sensors, 19(11), 2435.
--- Hatamie, A., Rahmati, R., Rezvani, E., Angizi, S., & Simchi, A. (2019). Yttrium hexacyanoferrate microflowers on freestanding three-dimensional graphene substrates for ascorbic acid detection. ACS Applied Nano Materials, 2(4), 2212-2221.
They were added, see page 3 lines 79-94.
- please add more information: “electropolymerization of a mononuclear Mn (II) complex” about the mechanism? reactions?
The term electropolymerization was used accidentally a mistake. The correct term is film of manganese complex. It was corrected, see page 4 lines 116-117. It is probably monomeric, see page 8-9 lines 225-257.
- page 10, line 370, “ Recovery rates of 100 ng L−1 vitamin C in the 95-105 % range” It is the very low concentration that has been used for recovery studies, please check the concentration units, linear range? it is not smaller than the linear range. Please see, Page 11, line 387, regression equation, and page 11, line 400.
It was accidentally wrong and corrected, i.e 100 μg L-1.
- Figure 4 and Figure 5, the current intensity is negative and then is positive ?!
In Figure 4 the absolute value of the reduction current intensity was used, which is usually the value that is countable for the manipulation of the experimental result in the case of reduction, while in figure 5 the real value of reduction peak current intensity is shown, see page 15 line 412-413.
- please add more information about electrocatalytic ascorbic acid detection at the modified electrodes. and interaction with ascorbic acid of electrode modifier. Figure 5,
see lines 314-393, 3.3 Electrochemical behavior of vitamin C on the modified Mn-GrOPE.
- page 5, line 219: “geometric area”
I do not agree with you about the calculation of geometric area of the electrode, in fact, in the purposed electrochemical method you can measure electroactive surface area, please check the "Bard books" and related references.
the geometric area is a physical area; you can see it. but the electrochemical reaction does not occur on all physical surfaces. for this reason, the estimated area is lower than the physical area.
It was corrected, see pages 10 lines 272-277.
- page 4, line 154, “ (0.1 mol L−1 acetate buffer pH 5.4 containing 0.008 mol L−1 KBr),” why you added KBr to buffer ???!
KBr was added because it was found amongst other salts that it increases the sensitivity of the detection; data are not shown.

Reviewer 3 Report
This work presents an electrochemical sensor for the detection of ascorbic acid. Although ascorbic acid has been extensively studied, I still think this paper has potential for publication. Please address the following questions.
1. Figures 4 and 6 suggest using professional graphing software, not excel.
2. What are the advantages of this sensor over other ascorbic acid sensors?
3. The authors could make a table comparing their sensor with the recently published sensors.
4. The authors have done very systematic anti-interference tests, but need to explain in which actual samples these anti-interference substances are present.
5. Why there are some curves with jitter in Figure 5?
Author Response
- Figures 4 and 6 suggest using professional graphing software, not excel.
It is derived from excel and modified using professional graphing software to be more presentable. It was corrected, see lines 411-414 and 425-427.
- What are the advantages of this sensor over other ascorbic acid sensors?
see page 4 lines 120-126.
- The authors could make a table comparing their sensor with the recently published sensors.
Table was made, see line 455.
- The authors have done very systematic anti-interference tests, but need to explain in which actual samples these anti-interference substances are present.
See page 4 line 122-124.
- Why there are some curves with jitter in Figure 5?
It is noise either from the instrument or from the GROPE electrode that couldn’t be removed even after the treatment with using the Savitzky and Golay filter (level 2) of the GPES software, followed by the GPES software moving average base-line correction, but it doesn’t interfere to the measurements.

Round 2
Reviewer 2 Report
The new version has been improved.
it can be accepted now.
Reviewer 3 Report
The revised version can be accepted.